# Anti-Cancer Effect of Cordycepin on FGF9-Induced Testicular Tumorigenesis

**DOI:** 10.3390/ijms21218336

**Published:** 2020-11-06

**Authors:** Ming-Min Chang, Siou-Ying Hong, Shang-Hsun Yang, Chia-Ching Wu, Chia-Yih Wang, Bu-Miin Huang

**Affiliations:** 1Department of Cell Biology and Anatomy, College of Medicine, National Cheng Kung University, Tainan 70101, Taiwan; hanxin.mmc@gmail.com (M.-M.C.); yingyinghong0120@gmail.com (S.-Y.H.); joshccwu@mail.ncku.edu.tw (C.-C.W.); 2Department of Physiology, College of Medicine, National Cheng Kung University, Tainan 70101, Taiwan; syang@mail.ncku.edu.tw; 3Department of Medical Research, China Medical University Hospital, China Medical University, Taichung 40400, Taiwan

**Keywords:** cordycepin, FGF9, cell proliferation, cell cycle, MA-10 mouse Leydig tumor cells, testicular cancer

## Abstract

Cordycepin, a bioactive constituent from the fungus *Cordyceps sinensis*, could inhibit cancer cell proliferation and promote cell death via induction of cell cycle arrest, apoptosis and autophagy. Our novel finding from microarray analysis of cordycepin-treated MA-10 mouse Leydig tumor cells is that cordycepin down-regulated the mRNA levels of *FGF9, FGF18, FGFR2* and *FGFR3* genes in MA-10 cells. Meanwhile, the IPA-MAP pathway prediction result showed that cordycepin inhibited MA-10 cell proliferation by suppressing FGFs/FGFRs pathways. The in vitro study further revealed that cordycepin decreased FGF9-induced MA-10 cell proliferation by inhibiting the expressions of p-ERK1/2, p-Rb and E2F1, and subsequently reducing the expressions of cyclins and CDKs. In addition, a mouse allograft model was performed by intratumoral injection of FGF9 and/or intraperitoneal injection of cordycepin to MA-10-tumor bearing C57BL/6J mice. Results showed that FGF9-induced tumor growth in cordycepin-treated mice was significantly smaller than that in a PBS-treated control group. Furthermore, cordycepin decreased FGF9-induced FGFR1-4 protein expressions in vitro and in vivo. In summary, cordycepin inhibited FGF9-induced testicular tumor growth by suppressing the ERK1/2, Rb/E2F1, cell cycle pathways, and the expressions of FGFR1-4 proteins, suggesting that cordycepin can be used as a novel anticancer drug for testicular cancers.

## 1. Introduction

Testicular Leydig cell tumors are the most common non-germ cell gonadal tumors although they overall account for 1–3% of testicular tumors, and usually occur in prepubertal boys (between 5 and 10 years) and men between 30 and 60 years [1,2,3]. This type of tumor is usually accompanied by an abnormal hormone secretion causing feminizing or virilizing syndromes [2,4]. Clinically, Leydig cell tumors are primarily treated by radical orchiectomy [5]. Testis sparing surgery has been used increasingly in both the adult and pediatric populations in order to maintain fertility [6]. Therefore, the focus of therapeutic strategies in testicular cancer is to avoid long term toxicity. The combination of two or more therapeutic treatments, such as surgery, chemotherapy and radiation therapy, is a more effective approach to specifically target cancer than a single therapy approach [7,8].

Cordycepin (3′-deoxyadenosine), an adenosine analogue, is a major bioactive component of the fungus *Cordyceps sinensis*, and has been suggested to exhibit many biological benefits and pharmacological activities, including immunomodulatory, anti-hyperglycemic, anti-fatigue, anti-inflammatory, anti-oxidative, neuroprotective, steroidogenesis, anti-microbial and anti-cancer activities [9,10]. Recently, cordycepin has been of great interest in medicinal applications because of its natural origin with anti-cancer properties and lesser side effects [10,11,12,13]. Cancer is a malignant disease characterized by the uncontrolled growth of abnormal cells, involved in the course of cell cycle progression and cell death (apoptosis), with the potential to metastasize [14,15]. Therefore, the majority of anticancer drugs exert their effects through the mechanisms of cell cycle arrest and the induction of apoptosis [14]. Numerous studies have showed that cordycepin inhibits cancer cells through not only inducing cell apoptosis and autophagy, but also through controlling cell proliferation. Several signaling pathways have been proposed for the apoptotic effects of cordycepin, including the activation of the death receptor 3 (DR3)/caspase signaling pathway [16,17,18], stimulation of adenosine receptors (ARs)/caspase 7/poly adenosine-diphosphate-ribose polymerase (PARP) pathway [19,20], by increasing the level of tumor necrosis factor-α (TNF-α)/tumor necrosis factor receptor (TNFR)/Bcl-2 signaling pathway [21,22], and by decreasing the phosphorylation levels of epidermal growth factor receptor (EGFR), AKT and ERK1/2 signaling pathway [23,24].

In the controlling of cell cycle, cordycepin enhances sub G1 accumulation in lung cancer [23,25], oral cancer [24] and testicular cancer [26,27], and induces G2/M arrest in bladder cancer [21,28] and colon cancer [29]. It was demonstrated that cordycepin inhibited the proliferation of B16-BL6 mouse melanoma cells by stimulating adenosine A3 receptors followed by glycogen synthase kinase (GSK)-3β activation and cyclin D1 inhibition [30]. Cordycepin can also inhibit the proliferation of H1657 lung cancer cells by decreasing EGFR phosphorylation and its downstream AKT and ERK1/2 signaling pathway [23]. Furthermore, cordycepin inhibits tumor growth and malignant transformation via inhibition of EGFR and IL-17RA-signaling in an oral cancer mouse model [24].

Recently, we reported that cordycepin induces MA-10 cell death by reducing FoxO/P15/P27/CDK4 pathways to arrest cell cycles and induce ER stress to cause UPR-dependent apoptosis through activating the PERK/eIF2α/ATF3/CHOP and the IRE1/XBP1 pathways [26]. To delineate the precise mechanisms underlying the anti-cancer effects of cordycepin in MA-10 cells, mRNA microarray analysis was performed on MA-10 cells treated with either 100 μM cordycepin or vehicle by using an Ingenuity Pathways Analysis (IPA) program [26]. Very interestingly, the fibroblast growth factors (FGFs) and fibroblast growth factor receptors (FGFRs) signaling pathways were identified and the RNA levels of FGF9, FGF18, FGFR2 and FGFR3 were significantly reduced in the cordycepin-treated group (Table 1). Meanwhile, the IPA-MAP prediction results showed that cordycepin inhibited the FGF signaling pathway through FGFRs to reduce cell growth (Figure 1). We previously also reported that FGF9 bound and activated FGFR2 to promote the ERK1/2 and Rb/E2F1 pathways, and subsequently increased the expression of cyclins and CDKs to enhance cell proliferation and tumor growth both in vitro and in vivo [31].

Fibroblast growth factor 9 (FGF9), a member of the FGF family, is a secreted and glycosylated 26 kDa protein that exerts mitogenic effects on different cell types, which is involved in various biological processes. In addition to embryo development [32,33], cell growth, morphogenesis [34], angiogenesis and tissue repair [35,36], FGF9 is essential for sex-determination and testicular development. Loss of FGF9 led to male-to-female sex-reversal, the reproductive system phenotypes in FGF9 knockout mice range from testicular hypoplasia to complete sex-reversal in females [37]. In addition, FGF9 is predominately expressed in the cytoplasm of Leydig cells of normal testis [38] and stimulates steroidogenesis in postnatal Leydig cells of testis [39]. Moreover, the FGF9 expression is significantly decreased in Sertoli cell-only syndrome (SCOS) patients [38]. Aberrant activation of FGF9 signaling pathways impacts not only biological processes but also carcinogenicity. Many studies have shown that FGF9 exerts oncogenic activity and promotes the neoplastic transformation, cell proliferation and invasive/metastatic behavior of lung cancer [40,41], gastric cancer [42,43,44], ovarian cancer [45,46], prostate cancer [47,48,49] and testicular cancer [31]. Clinically, high expression of FGF9 is associated with poor prognosis in patients who have non-small cell lung cancer (NSCLC) [50].

In the present study, we demonstrated that cordycepin could significantly inhibit FGF9-induced MA-10 cell tumorigenesis in vitro and in vivo by the suppression of MAPK and Rb/E2F1 pathways, and the downregulation of cell cycle proteins and FGFRs, implying that cordycepin may be used as a new drug for testicular cancer therapy.

## 2. Results

### 2.1. Cordycepin Inhibited FGF9-Induced Cell Viability of MA-10 Cells

To investigate the effects of cordycepin on FGF9-induced cell proliferation in MA-10 cells, microscopy examination and trypan blue exclusion assays were performed. The results showed that the number of rounded-up MA-10 cells progressively increased as the dose of cordycepin increased, whereas MA-10 cells firmly attached with spindle shapes in control group (0 μM cordycepin) (Figure 2A). The cell proliferation rates of FGF9-treated MA-10 cells were significantly decreased dose-dependently in cordycepin-treated group at 24 h after treatment (Figure 2B). In addition, the cell proliferation rate of MA-10 cells without FGF9 exposure was also suppressed by cordycepin (Figure 2A,B). No significant difference of cell proliferation rates was observed between FGF9-exposed cells and unexposed cells after treating with 50, 100 and 200 µM of cordycepin, indicating that cordycepin could inhibit FGF9-induced cell proliferation in MA-10 cells.

### 2.2. Cordycepin Inhibited FGF9-Induced ERK1/2 and pRb/E2F Pathway in MA-10 Cells

We next investigated whether cordycepin could suppress the signaling pathway induced by FGF9 in MA-10 cells. The results showed that FGF9-induced phosphor-ERK1/2 (p-ERK1/2) expression was significantly inhibited by cordycepin at 0.25 and 12 h after treatment (Figure 3A). At 24 h after FGF9 treatment, the phosphorylation of ERK1/2 was not elevated. However, the basal protein levels of p-ERK1/2 were significantly reduced by cordycepin (Figure 3A). The effects of cordycepin on the p-Rb/E2F pathway and the downstream signaling of ERK1/2 were also examined. Cordycepin (25, 50 and 100 µM) significantly inhibited FGF9-induced phosphorylation of Rb at 0.25 and 12 h, but not at 24 h after treatments (Figure 3B), and also inhibited FGF9-induced E2F1 expression 12 h after treatments (Figure 3C). These data indicated that cordycepin could inhibit FGF9-induced Rb phosphorylation and E2F1 overexpression, and subsequently suppress cell proliferation in MA-10 cells.

### 2.3. Cordycepin Reduced the Expression of Cyclins and CDKs in FGF9-Treated MA-10 Cells

According to our previous study, which showed that FGF9 did increase the expressions of cyclins and CDKs to promote cell cycle progression for MA-10 cell proliferation [31], the effects of cordycepin on cell cycle progression in FGF9-treated MA-10 cells were investigated. Consistent with previous data [31], FGF9 could induce cyclin D1, cyclin E1 and cyclin A1 at 12 h after treatment (Figure 4A–D), and up-regulate cyclin B1 at 24 h after treatment (Figure 4A,E). In the 12 h FGF9-treated group, the FGF9-induced overexpression of cyclin D1, cyclin E1 and cyclin A1 could be reversed by cordycepin in a dose-dependent manner (Figure 4A–D), whereas the expression of cyclin B1, had not yet been induced by FGF9 and was also down-regulated by cordycepin (Figure 4A,E). In the 12 h control group, the expression of cyclin A1 and cyclin B1 were also significantly reduced by cordycepin (Figure 4A,D,E). At 24 h after treatment, FGF9-induced cyclin B1 could be significantly suppressed by 100 µM cordycepin (Figure 4A,E). In addition, cordycepin did reduce protein basal levels of cyclin B1 and E1 proteins whether treated with FGF9 or not at 12 h after treatment (Figure 4A,D,E). These data illustrated that cordycepin could affect cell cycle progression by downregulating cyclin D1, cyclin E1, cyclin A1 and cyclin B1 proteins in FGF9-treated MA-10 cells.

Next, we examined whether cordycepin could affect the FGF9-induced expression of different CDK proteins. Again, consistent with previous findings [31], FGF9 significantly induced CDK4, CDK 2 and CDK1 protein expressions at 12 h, but not at 0.25, after treatment (Figure 5A–D). Cordycepin had no effects on CDKs at 0.25 h after treatments. But, at 12 h after FGF9 treatment, FGF9-induced expressions of CDK4 was suppressed by cordycepin in a dose-dependent manner (Figure 5A,B), whereas FGF9-induced expressions of CDK2 and CDK1 were suppressed by cordycepin at the dose of 50 and 100 µM (Figure 5A,C,D). These results indicated that cordycepin could inhibit FGF9-induced expressions of cyclins and CDKs in MA-10 cells, which could prevent cell cycle progression by inhibiting the FGF9-induced cell cycle-related protein expressions in MA-10 cells.

### 2.4. Cordycepin Decreased FGF9-Induced FGFR1-4 Expressions in MA-10 Cells

In mammals, FGFs bind and activate FGFR to trigger signaling in a variety of biological processes [51]. Although different FGFs have different affinities for FGFRs, in general, most FGFs can bind with any of the four major FGFRs [52]. In many studies, some cancer developments and progressions are closely related to the overexpression of FGFRs [53,54]. We have reported that FGF9 promoted the expressions of FGFR1, FGFR2, FGFR3 and FGFR4 at 24 h after treatment [31]. Microarray analysis also indicated that cordycepin inhibited FGFR2 and FGFR3 mRNA levels in MA-10 cells at 3 h after treatment. To confirm the inhibitory effects of cordycepin on the expression of FGFRs, the protein levels of FGFRs in FGF9-treated MA-10 cells were determined by western blot analysis. At 72 h after treatment, cordycepin did significantly and dose-dependently reduce FGF9-induced FGFR1, FGFR2, FGFR3 and FGFR4 expressions, respectively (Figure 6A–E). Moreover, cordycepin exerted an inhibitory effect on FGFRs at a high dose of 100 μM in FGF9-untreated MA-10 cells (Figure 6A–E).

### 2.5. Cordycepin Inhibited FGF9-Promoted Testicular Tumor Growth In Vivo

We next used a C57BL/6J mouse model to confirm the anti-cancer effect of cordycepin on FGF9-induced testicular tumorigenesis. MA-10 cells were subcutaneously injected in the right back of mice. Seven days after tumor inoculation, mice were randomly assigned to receive the intratumoral (i.t.) injection of 5 ng FGF9, BSA (the vehicle control for FGF9) or no treatment (Control), and intraperitoneal (i.p.) injection of 20 mg/kg cordycepin or PBS (the vehicle control for cordycepin). On day 9 after initiation of drug administration, the tumor volumes of the FGF9/PBS group were significantly increased compared to the BSA/PBS and Control/PBS groups (Figure 7A), which was consistent with our previous finding in a NOD-SCID model [31]. In both FGF9/cordycepin and Control/cordycepin groups, cordycepin treatment resulted in significantly smaller tumor volume compared to FGF9/PBS and Control/PBS groups, respectively (Figure 7A). However, there are no significantly differences between Control/cordycepin and FGF9/cordycepin (Figure 7A). The same pattern can be observed in the size and weight of tumors (Figure 7B,C). These data indicated that cordycepin could inhibit FGF9-induced Leydig cell tumor growth in vivo.

To further investigate the inhibitory effect of cordycepin on FGF9-induced tumor in vivo, immunohistochemistry (IHC) examinations of Ki-67, cleaved caspase-3 and FGF9 were carried out. Results showed that the tumor tissue expression of Ki-67 decreased and cleaved caspase-3 increased in FGF9/cordycepin and Control/cordycepin groups (Figure 7D–G), whereas the tumor tissue expression of Ki-67 increased and cleaved as caspase-3 decreased in the FGF9/PBS group (Figure 7D–G). These data indicated that, in vivo, cordycepin decreased the tumor mass by decreasing MA-10 cell proliferation and increasing cell apoptosis.

Furthermore, the immunohistochemical expressions of p-ERK1/2 and p-Rb in MA-10 tumors were significantly inhibited in FGF9/cordycepin and Control/cordycepin groups while the p-ERK1/2 and p-Rb expressions were increased in the FGF9/PBS group (Figure 8A–D). These in vivo observations further support that cordycepin inhibited FGF9-induced cell proliferation by suppressing the ERK1/2 and RB/E2F pathways in vitro (Figure 3).

### 2.6. Cordycepin Inhibited the Expression of FGFR1-4 In Vivo

We further observed the effects of cordycepin on FGFR1-4 expressions when MA-10 tumor was exposed to FGF9 in vivo. The immunohistochemical expression of FGFR1-4 was increased in the FGF9/PBS group and decreased in the groups treated with cordycepin whether the tumor was exposed to FGF9 or not (Figure 9A–E). These data suggested that cordycepin could decrease the expression of FGFR1-4 in MA-10 cells both in vitro (Figure 6) and in vivo, implying that cordycepin exhibited potential therapeutic characteristics by suppressing FGF/FGFRs signaling pathways in MA-10 cells.

## 3. Discussion

In the present study, we have a novel finding that cordycepin inhibited MA-10 cell proliferation in vitro and tumor growth in vivo through suppressing the FGF9 signaling pathway, which upregulated the phosphorylation of ERK1/2 and Rb, the expressions of E2F1 and the cell cycle related proteins, cyclins and CDKs (Figure 10).

Cordycepin has important clinical implications due to its significant antiviral and anticancer activities. In recent decades, numerous reports have indicated that cordycepin has remarkable anticancer potential, such as inhibition of cell proliferation, induction of apoptosis, and inhibition of cell migration, tumor growth and invasiveness [9,10]. These inhibitory effects on tumors have been observed in oral cancer [55], lung cancer [22], gastric cancer [56], pancreatic cancer [57], prostate cancer [58], testicular cancer [26,27] and colorectal carcinoma [18,59]. The mechanisms underlying the anticancer effects of cordycepin have been intensively investigated. However, very few studies have been investigated regarding any effects between cordycepin and FGFs upon tumorigenesis and/or any biomedical related events. Sun el al. reported that administration of *Cordyceps militaris* decreased the concentration and the activity of cytokines, especially inflammatory cytokines and chemokines, such as eotaxin, FGF2, EGF, VEGF, G-CSF, GM-CSF, IL-8, IL-10 TNF-α and interferon-γ, in blood samples from healthy volunteers [60].

Here, from the microarray analysis of cordycepin-mediated alteration of gene expression levels in testicular cancer cells, the clustering of the microarray data identified that the mRNA level of FGF9 and FGF18 were reduced in 100 μg/mL cordycepin-treated MA-10 cells (Table 1). Meanwhile, the animal experiment results showed that the IHC expression of FGF9 in FGF9-treated MA-10 tumor tissues was reduced by cordycepin (Figure 7D). These findings implied that cordycepin is involved in regulating the secretion of growth factors and/or cytokines.

Previously, we demonstrated that FGF9 signaling through FGFR2-stimulated cell proliferation and tumor growth in MA-10 cells [31]. FGFs bind four high affinity ligand-dependent FGF receptors (FGFR1–4), and the downstream signaling of FGF/FGFR pathways include PLCγ/PKC, STAT, PI3K/AKT and Ras/Raf/MEK/MAPK [31,61,62]. FGFs and FGFRs would alter in cancer cells associated with receptor overexpression and/or the aberrant FGFs secretion [61]. FGF9 has been reported to exhibit the high affinity with FGFR2 and FGFR3, and the low affinity to FGFR1 and FGFR4 [63,64,65,66]. In our observations, FGF9 stimulated the expression of FGFR1-4 in MA-10 tumor cells in vitro and in vivo [31] (Figure 6 and Figure 8). This may lead to enhance FGF/FGFR signaling activities which make the way for FGF9 to promote MA-10 tumor growth, since we observed that FGF9 activated the downstream signaling through FGFR2 in MA-10 cells [31]. In addition, we found that cordycepin exerted the anti-tumor activity on MA-10 tumors both in severe combined immunodeficiency (SCID) mouse [31] and C57BL/6J mouse models (Figure 7). The expressions of FGFR1-4 in MA-10 cells and inoculated MA-10 tumors were reduced after cordycepin exposure, indicating that cordycepin could reduce tumor growth through FGFR-signaling pathways.

We have previously demonstrated that cordycepin could stimulate steroidogenesis through PKA and PKC pathways in MA-10 cells [67]. We also illustrate that cordycepin could induce apoptosis through p38 MAPKs, PI3K/AKT, cell cycle arrest and caspase signaling pathways, and promote unfolded protein response-dependent cell death through FoxO/P15/P27, PERK-eIF2α and the IRE1-XBP1 pathways in MA-10 cells [26,27,68]. Moreover, FGF9 could activate the FGFR2 signaling pathway to induce MA-10 cell proliferation [31]. However, it remains unclear whether cordycepin would directly affect FGFR2 function to suppress FGF9-treated MA-10 cell proliferation. Or, indeed, cordycepin blocked FGFR2 downstream signaling pathway to suppress FGF9-treated MA-10 cell proliferation. In fact, our current results show that the expressions of all four FGF receptors (FGFR1-4) were significantly reduced by cordycepin in FGF9-treated MA-10 cells (Figure 6A–E). In addition, cordycepin did inhibit the expressions of all four FGFRs in MA-10 tumor tissues in vivo. Thus, it is highly possible that cordycepin would directly down-regulate FGFR2 function to further suppress FGF9-treated MA-10 cell proliferation. However, it should be noted that all results in the present study could only illustrate indirect evidence how cordycepin possibly affected FGF9/FGFR2 signaling pathways. It will be helpful to confirm that cordycepin does directly affect FGFR2 by using FGFR2-knockdown experiment in MA-10 cells, which should be valuable to further investigations in the near future.

It has been reported that the anti-cancer activities of cordycepin can be mediated by several putative receptors, such as adenosine receptors (ARs) [20,67,68], death receptors (DRs) [69,70], and the epidermal growth factor receptor (EGFR) [23,24]. Cordycepin can bind to the AR3 to inhibit cell proliferation by inactivating glycogen synthase kinase (GSK)-3β/β-catenin signaling pathway, and subsequently suppressing cyclin D1 and c-myc expression in melanoma and lung carcinoma cells [11,67,68]. Cordycepin could cause cell cycle arrest via the inhibition of the cyclin B/CDK complex at G2/M transition in human bladder cancer cell lines [28] or down-regulate cyclin E1, cyclin A and CDK2 to induce S phase arrest in human lung cancer cells and gallbladder cancer cells [71,72]. Similar results were observed in our study that FGF9-stimulated expression of cell cycle-related proteins, including cyclin D1, cyclin E1, cyclin A1, cyclin B1, CDK4, CDK2, and CDK1 [31], were down-regulated by cordycepin (Figure 4 and Figure 5). Data presented here and in previous studies indicate that cordycepin can inhibit many malignant tumors through different signaling pathways. Moreover, cordycepin, compared to other chemotherapeutic drugs, was low toxic to normal cells and mouse normal tissues when inhibiting the growth and progression of cancer cells in vitro and in vivo [9,13]. Therefore, cordycepin may be considered as an ideal drug candidate for cancer treatment.

In conclusion, our studies revealed the anti-cancer effect of cordycepin on FGF9-induced testicular tumor growth by affecting the expressions of p-ERK1/2, p-Rb, E2F1, cell cycle related proteins, and FGFR1-4 proteins. Cordycepin, the bioactive compound isolated from Chinese herb, might be used as a novel anticancer drug for testicular and/or other cancers.

## 4. Materials and Methods

### 4.1. Cell Lines and Treatments

The MA-10 mouse Leydig tumor cell line was a gift from Dr. Mario Ascoli (Department of Pharmacology, University of Iowa, IA, USA). MA-10 cells were maintained in modified Waymouth’s MB 752/1 medium containing 20 mM HEPES, 1.12 g/L NaHCO_3_ and supplemented 10% fetal bovine serum (FBS). 3 × 10^5^ MA-10 cells were seeded in 60-mm dishes. After 19 h serum starvation, cells were treated with FGF9 (50 ng/mL) and/or cordycepin (0, 25, 50, 100 and 200 μM) in the medium containing 1% FBS for 0.25, 12 and 24 h. All chemicals and materials used in this study were listed in Appendix A.

### 4.2. Cell Proliferation Assay

MA-10 cells were seeded in 60 mm dishes at a density of 2 × 10^5^ cells/dish. After 19 h serum-free starvation, the cells were treated with/without 50 ng/mL FGF9 and/or different concentrations of cordycepin in medium containing 1% FBS for 24 h. Then, treated cells were suspended by trypsin and mixed with an equal volume of 0.4% trypan blue solution. The stained (dead) and unstained (live) cells were counted on a hemocytometer, and the cell viability was computed as the percentage of live cells in the sample. Cell proliferation rates were evaluated by comparing the cell viability between the treatment and control groups.

### 4.3. Western Blot Analysis

Treated MA-10 cells were washed with ice-cold PBS and lysed by ice-cold lysis buffer containing 20 mM Tris pH 7.5, 150 mM NaCl, 1 mM EDTA, 1 mMEGTA, 1% Triton X-100, 2.5 nM sodium pyrophosphate and 1 mM sodium orthovanadate. After centrifugation (12000× *g*, 12 min, 4 °C), supernatants were collected and stored at −80 °C until further analysis. 25 μg total protein was separated by 10% sodium dodecyl sulfate polyacrylamide gel electrophoresis (SDS-PAGE) and transferred onto polyvinylidene difluoride (PVDF) membranes. The PVDF membranes with transferred protein were blocked with 5% nonfat milk for 1 h and then incubated with the primary antibody for overnight at 4 °C. After being incubated with horseradish peroxidase (HRP)-conjugated secondary antibody, the signal was visualized with chemiluminescence HRP substrate by using a computer-assisted image analysis system (UVP bioimage system software, UVP Inc., CA, USA). The integrated optical density (IOD) were quantitated by using ImageJ software (NIH, Bethesda, USA). Protein levels were normalized to β-actin and expressed as fold change by comparison to the control group at each time point. All antibodies used in this study were listed in Appendix A.

### 4.4. Animals and Treatments

Six-week-old male C57BL/6J mice were purchased from National Cheng Kung University (NCKU) Animal Center (Tainan, Taiwan). Mice received a subcutaneous (SC) injection of 7.5 × 10^5^ MA-10 cells in the right hip. Seven days after tumor inoculation, tumor-bearing mice were randomly assigned to five groups (Control/PBS, FGF9/PBS, BSA/PBS, FGF9/codycepin and Control/cordycepin groups; 5 mice per group), and each mouse received an intratumoral (i.t.) injection of FGF9 or BSA and an intraperitoneal (i.p.) injection of cordycepin or PBS daily (i.t./i.p.). In the control group, mice were inoculated with MA-10 cells and received no i.t. injection. In the FGF9 group, 100 μL of 50 ng/mL FGF9 was locally injected into the MA-10 cell-forming tumor mass, whereas 100 µL of 0.0025% BSA were injected into MA-10 tumor mass as the vehicle control of FGF9. FGF9 injecting solution was freshly prepared daily by diluting FGF9 stock solution (20 μg/mL FGF9 in PBS with 0.1% BSA). Cordycepin (25 mg/kg/day) was administered by i.p. injection and the cordycepin control group received the same volume of PBS. Tumor size and body weight were measured daily. When mice were sacrificed, MA-10 tumor tissues were collected, weighted and fixed by a 4% paraformaldehyde solution. Tumor volume was calculated by the formula: 0.52 × length × width × width [73]. All experiments were performed in accordance with relevant guidelines and regulations, which were approved by the Institutional Animal Care and Use Committee (IACUC) of NCKU (IACUC approval no.: 105102; approval date: 2015.12.23.).

### 4.5. Immunohistochemistry (IHC)

Formalin-fixed paraffin-embedded tumor tissue sections were dewaxed in xylene, dehydrated in ethanol, and washed in PBS. Endogenous peroxidase activity was blocked by incubation with 0.3% H_2_O_2_ followed by washing with PBS. Sodium citrate buffer (10 mM sodium citrate and 0.05% tween 20, pH6.0) was used to retrieve the antigen for 50 min at 120 °C autoclave. The sections were blocked with 2% nonfat milk for 1 h and were incubated overnight at 4 °C with primary antibodies. Signal was visualized using HRP-conjugated secondary antibody and the chromogenic substrate 3,3′-diaminobenzidine (DAB). The sections were then counterstained with hematoxylin, dehydrated with a graded ethanol series, cleared in xylene, and mounted with a coverslip using a mounting solution. Negative controls were performed in each IHC assay by replacing the primary antibodies with a corresponding non-specific IgG. All antibodies used in this study were listed in Appendix A.

### 4.6. Statistical Analysis

All statistical analysis were carried out using GraphPad Prism 6 software (GraphPad, La Jolla, CA, USA). *p* < 0.05 was considered to be statistically significant in this study.

## Figures and Tables

**Figure 1 ijms-21-08336-f001:**
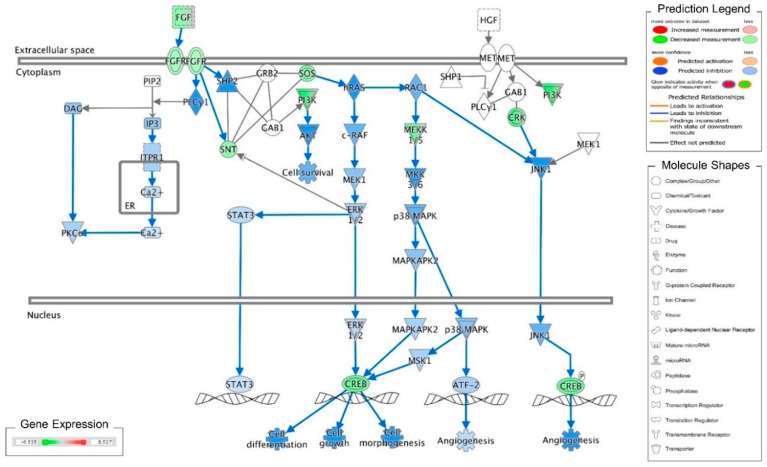
Ingenuity pathway analysis of FGF/FGFR pathway suppressed by cordycepin in MA-10 cells. Microarray analysis was performed on MA-10 cells treated with either 100 μM cordycepin or a vehicle for 3 h. The genes significantly up- and down-regulated by cordycepin were overlaid onto the canonical signaling pathways by using an Ingenuity Pathways Analysis (IPA) program (IPA, QIAGEN Inc., https://www.qiagenbioinformatics.com/products/ingenuity-pathway-analysis). The genes significantly regulated by cordycepin were mapped onto the canonical FGF signaling pathway. The activation or inhibition effects of the pathways were predicted by the molecule activity predictor (MAP) overlay tools in the IPA program. Red-Green color indicates the level of gene expression. Blue color indicates the inhibitory effect, whereas an orange color indicates an activating effect. Yellow color means that the finding is inconsistent with the state of downstream molecules, and a gray color means no prediction on the path.

**Figure 2 ijms-21-08336-f002:**
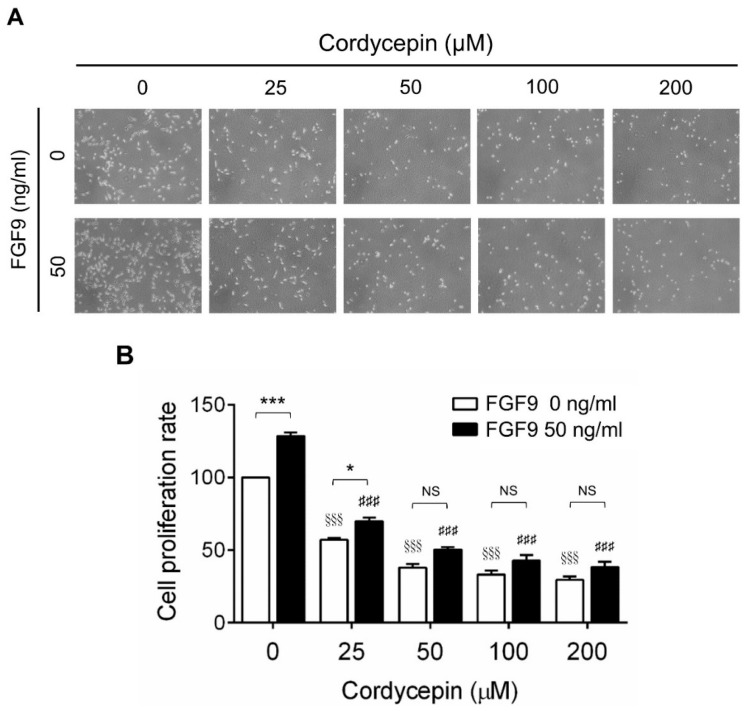
Inhibition of the FGF9-induced cell proliferation of MA-10 cells by cordycepin. (**A**) Morphology observations of MA-10 cells under an optical microscope (400×) after treatment without or with FGF9 (50 ng/mL) and/or different concentrations of cordycepin (0, 25, 50, 100 and 200 μM) for 24 h. (**B**) Trypan blue exclusion assay of MA-10 cell proliferation rates after treatment. Quantitative values are shown as the mean ± SEM, *n* = 4. *p* values were calculated using two-way ANOVA with Tukey’s multiple comparisons post-tests. * *p* < 0.05, *** *p* < 0.001 compared to the control group (0 ng/mL FGF9) at each dose of cordycepin; ^§§§^
*p* < 0.001 compared to the group with 0 μM cordycepin and 0 ng/mL FGF9 treatments; ### *p* < 0.001 compared to the group with 0 μM cordycepin and 50 ng/mL FGF9 treatments.

**Figure 3 ijms-21-08336-f003:**
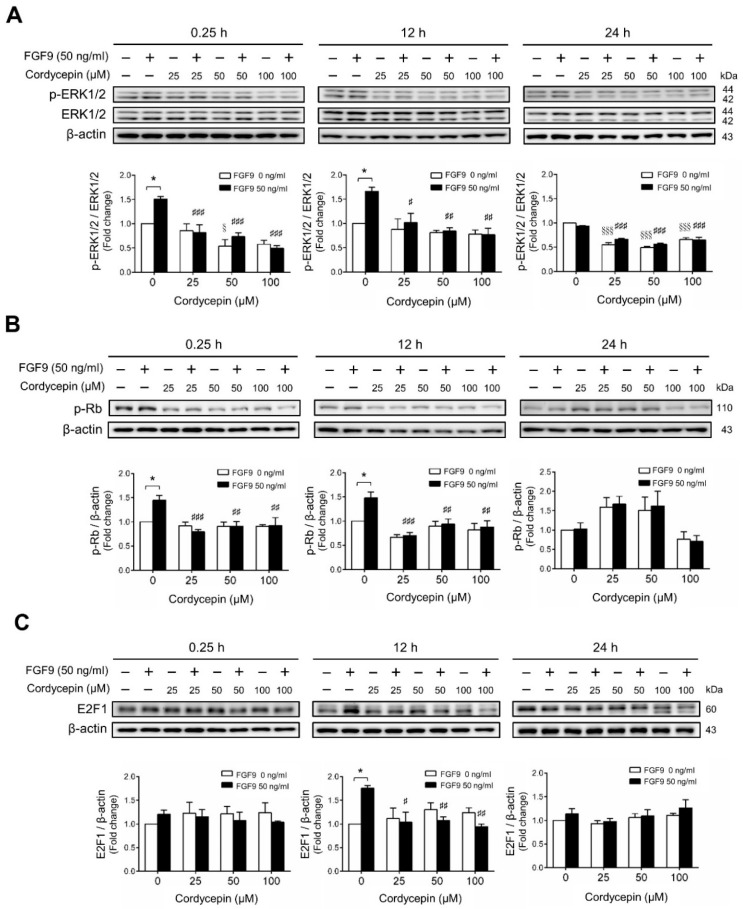
Cordycepin suppressed FGF9-induced expression of p-ERK1/2, p-Rb and E2F1 in MA-10 cells. Western blot analysis for the expression of (**A**) total ERK1/2, p-ERK1/2 (Thr202/Tyr204), (**B**) p-Rb and (**C**) E2F1 in MA-10 cells treated without or with FGF9 (50 ng/mL) and different concentrations of cordycepin (0, 25, 50 and 100 µM) for 0.25, 12 and 24 h, respectively. Quantitative analysis of Western blotting using ImageJ software. Values are shown as the mean ± SEM, *n* = 4. *p* values were calculated using two-way ANOVA with Tukey’s multiple comparisons post-tests. * *p* < 0.05 compared to the control group (0 ng/mL FGF9) at each dose of cordycepin; ^§§§^
*p* < 0.001 compared to the group with 0 μM cordycepin and 0 ng/mL FGF9 treatments; # *p* < 0.05, ## *p* < 0.01, ### *p* < 0.001 compared to the group with 0 μM cordycepin and 50 ng/mL FGF9 treatments.

**Figure 4 ijms-21-08336-f004:**
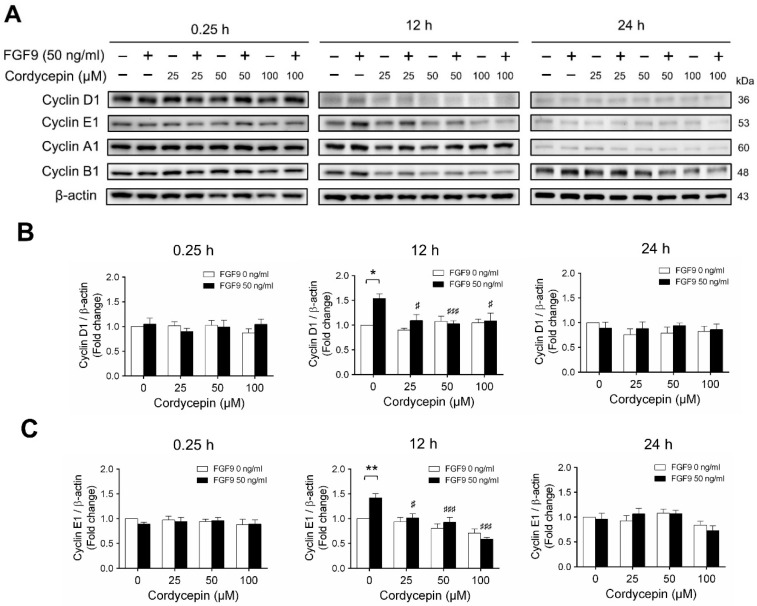
Cordycepin suppressed FGF9-induced expression of cyclin D1, cyclin E1, cyclin A1 and cyclin B1 in MA-10 cells. (**A**) Western blot analysis of cyclin D1, cyclin E1, cyclin A1 and cyclin B1 expression in MA-10 cells treated without or with FGF9 (50 ng/mL) and different concentrations of cordycepin (0, 25, 50 and 100 µM) for 0.25, 12 and 24 h. (**B**–**E**) The relative expression of (**B**) cycline D1, (**C**) cyclin E1, (**D**) cyclin A1 and (**E**) cyclin B1 was quantified using ImageJ software by normalization with β-actin. Values are shown as the mean ± SEM, *n* = 4. *p* values were calculated using two-way ANOVA with Tukey’s multiple comparisons post-tests. * *p* < 0.05, ** *p* < 0.01, compared to the control group (0 ng/mL FGF9) at each dose of cordycepin; ^§§^
*p* < 0.01, ^§§§^
*p* < 0.001 compared to the group with 0 μM cordycepin and 0 ng/mL FGF9 treatments; # *p* < 0.05, ### *p* < 0.001 compared to the group with 0 μM cordycepin and 50 ng/mL FGF9 treatments.

**Figure 5 ijms-21-08336-f005:**
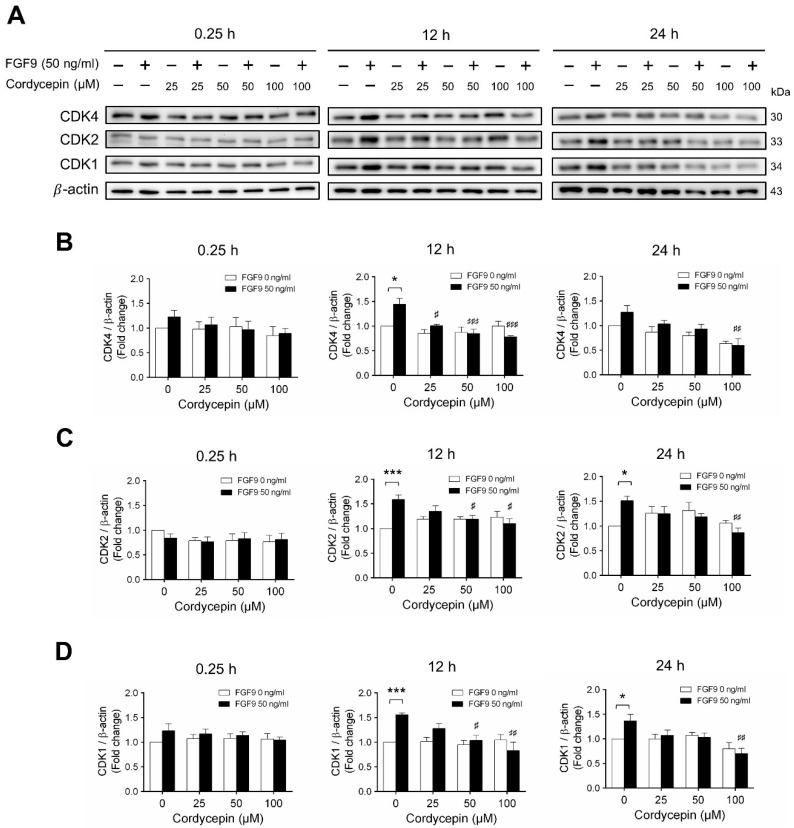
Cordycepin suppressed FGF9-induced expression of CDK4, CDK2 and CDK1 in MA-10 cells. (**A**) Western blot analysis for the expression of CDK4, CDK2 and CDK1 in MA-10 cells were treated without or with FGF9 (50 ng/mL) and different concentrations of cordycepin (0, 25, 50 and 100 µM) for 0.25, 12 and 24 h. (**B**–**D**) The relative expression of (**B**) CDK4, (**C**) CDK2, and (**D**) CDK1 was quantified using ImageJ software by normalization with β-actin. Values are shown as the mean ± SEM, *n* = 4. *p* values were calculated using two-way ANOVA with Tukey’s multiple comparisons post-tests. * *p* < 0.05, *** *p* < 0.001 compared to the control group (0 ng/mL FGF9) at each dose of cordycepin; # *p* < 0.05, ## *p* < 0.01, ### *p* < 0.001 compared to the group with 0 μM cordycepin and 50 ng/mL FGF9 treatments.

**Figure 6 ijms-21-08336-f006:**
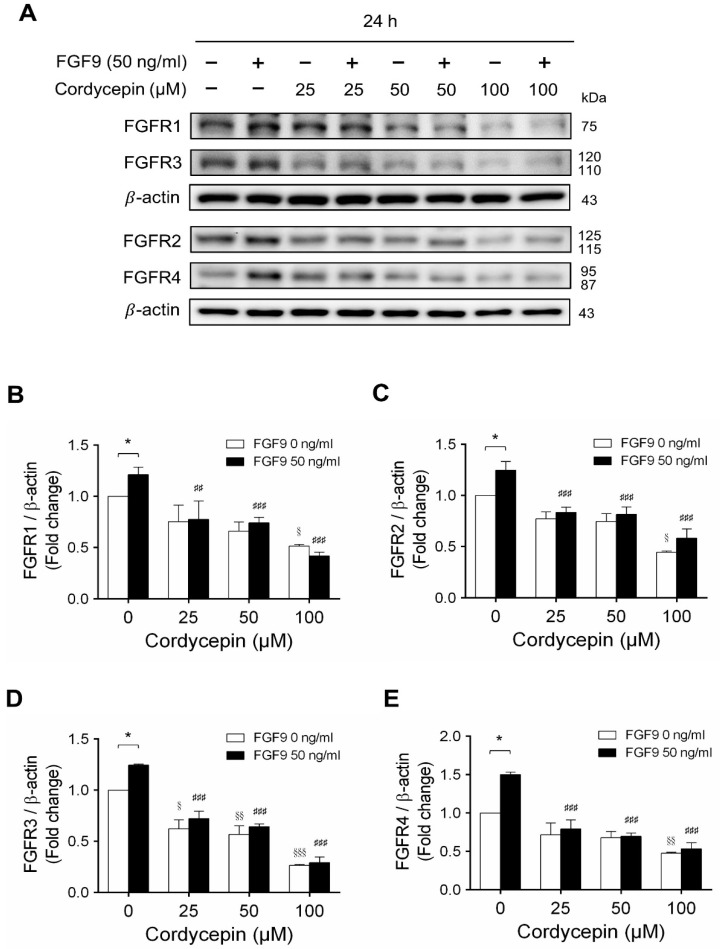
Cordycepin decreased FGF9-induced expressions of FGFR1-4 proteins in MA-10 cells. (**A**) Western blot analysis for the expression of FGFR1, FGFR2, FGFR3 and FGFR4 in MA-10 cells treated without or with FGF9 (50 ng/mL) and different concentrations of cordycepin (0, 25, 50 and 100 µM) for 12, 24, 48 and 72 h. (**B–E**) The relative expression of (**B**) FGFR1, (**C**) FGFR2, (**D**) FGFR3 and (**E**) FGFR4 was quantified using ImageJ software by normalization with β-actin. Values are shown as the mean ± SEM, *n* = 4. *p* values were calculated using two-way ANOVA with Tukey’s multiple comparisons post-tests. * *p* < 0.05, compared to the control group (0 ng/mL FGF9) at each dose of cordycepin; ^§^
*p* < 0.05, ^§§^
*p* < 0.01, ^§§§^
*p* < 0.001 compared to the group with 0 μM cordycepin and 0 ng/mL FGF9 treatments; ## *p* < 0.01, ### *p* < 0.001 compared to the group with 0 μM cordycepin and 50 ng/mL FGF9 treatments.

**Figure 7 ijms-21-08336-f007:**
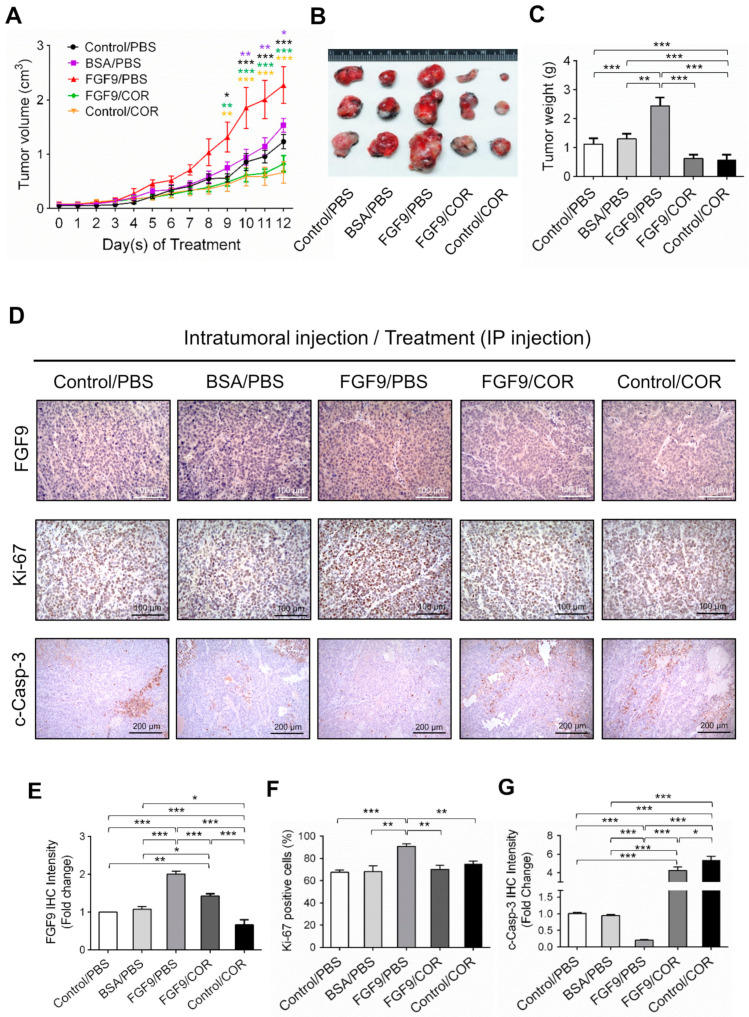
Cordycepin suppressed in vivo tumor growth in the C57BL/6J mouse model of FGF9-induced testicular tumorigenesis. Seven days after MA-10 cell inoculation, mice were randomly allocated to five groups (i.t./i.p.): Control/PBS, BSA/PBS, FGF9/PBS, FGF9/COR and Control/COR groups. In the control group, mice were inoculated with MA-10 cells and received no i.t. injection. (**A**) Tumor growth curves of the MA-10 tumor-bearing mice received different treatments, as indicated in the plots. (**B**) Photograph of excised MA-10 tumors and (**C**) tumor weights at the time mice were sacrificed. (**D**) The immunohistochemical (IHC) assay of the expression of FGF9, Ki-67 and cleaved caspase-3 (c-Casp-3) on MA-10 tumor tissue from Control/PBS, BSA/PBS, FGF9/PBS, FGF9/COR and Control/COR-treated mice (original magnification: FGF9 and Ki-67, x200; C-Casp-3, x100). (**E**–**G**) IHC quantification of the expression of (**E**) FGF9, (**F**) Ki-67 and (**G**) c-Casp-3 in MA-10 tumor tissue from (**D**). Quantification values are represented as the mean ± SEM, *n* = 5. *p* values were calculated using two-way ANOVA with Sidak’s multiple comparisons post-tests (**A**), **p* < 0.05, ** *p* < 0.01, *** *p* < 0.001 with different color indicates a statistically significant difference at each time point compared to their corresponding control group; and one-way ANOVA with Tukey’s multiple comparisons post-tests (**C**,**E**–**G**), * *p* < 0.05, ** *p* < 0.01, *** *p* < 0.001. COR: cordycepin.

**Figure 8 ijms-21-08336-f008:**
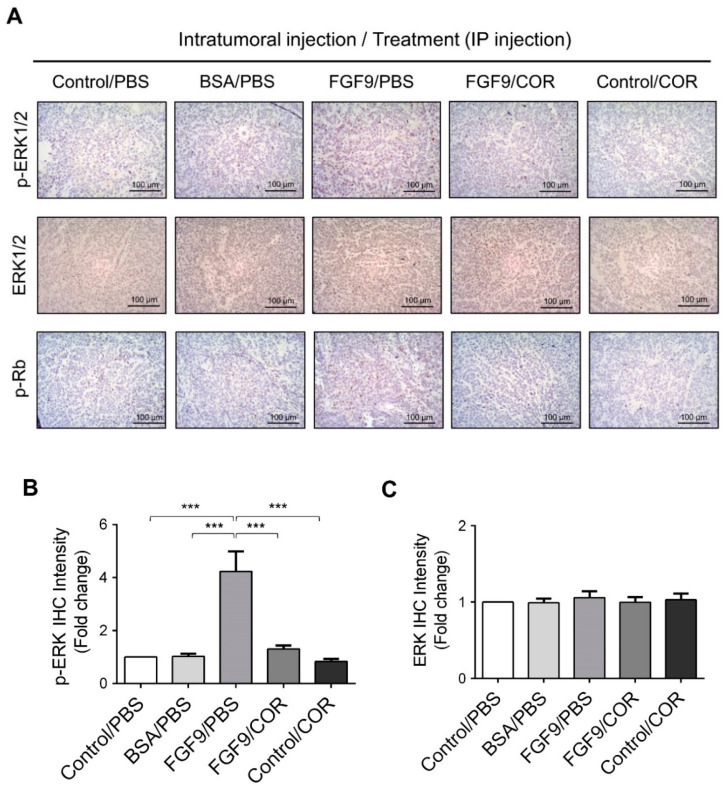
Cordycepin inhibited the phosphorylation of ERK1/2 and Rb in FGF9-induced allograft model. (**A**) The immunohistochemical (IHC) assay of the expression patterns of p-ERK1/2, total ERK1/2 and p-Rb in MA-10 tumor tissue from Control/PBS, BSA/PBS, FGF9/PBS, FGF9/COR and Control/COR treated mice (original magnification 200×). (**B**–**D**) IHC quantification of the expression of (**B**) pERK1/2, (**C**) total ERK1/2 and (**D**) p-Rb in MA-10 tumor tissue from (**A**). Quantification values are represented as the mean ± SEM, *n* = 5. *p* values were calculated using one-way ANOVA with Tukey’s multiple comparisons post-tests, *** *p* < 0.001. COR: cordycepin.

**Figure 9 ijms-21-08336-f009:**
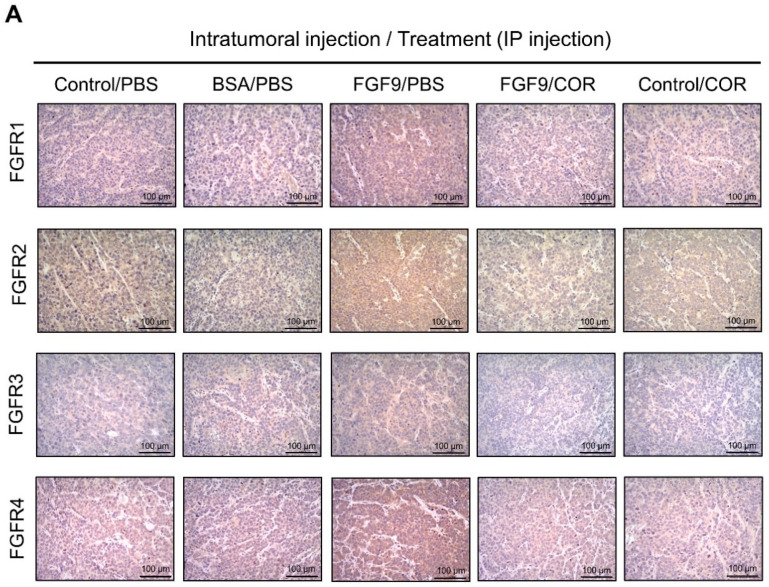
Cordycepin reduced the expression of FGFR1-4 in FGF9-induced allograft model. (**A**) The immunohistochemical (IHC) assay of the expression patterns of FGFR1, FGFR2, FGFR3 and FGFR4 in MA-10 tumor tissues from Control/PBS, BSA/PBS, FGF9/PBS, FGF9/COR and Control/COR treated mice (original magnification 200×). (**B**–**E**) IHC quantification of the expression of (**B**) FGFR1, (**C**) FGFR2, (**D**) FGFR3 and (**E**) FGFR4 in MA-10 tumor tissue from (**A**). Quantification values are represented as the mean ± SEM, *n* = 5. *p* values were calculated using one-way ANOVA with Tukey’s multiple comparisons post-tests, ** *p* < 0.01, *** *p* < 0.001. COR: cordycepin.

**Figure 10 ijms-21-08336-f010:**
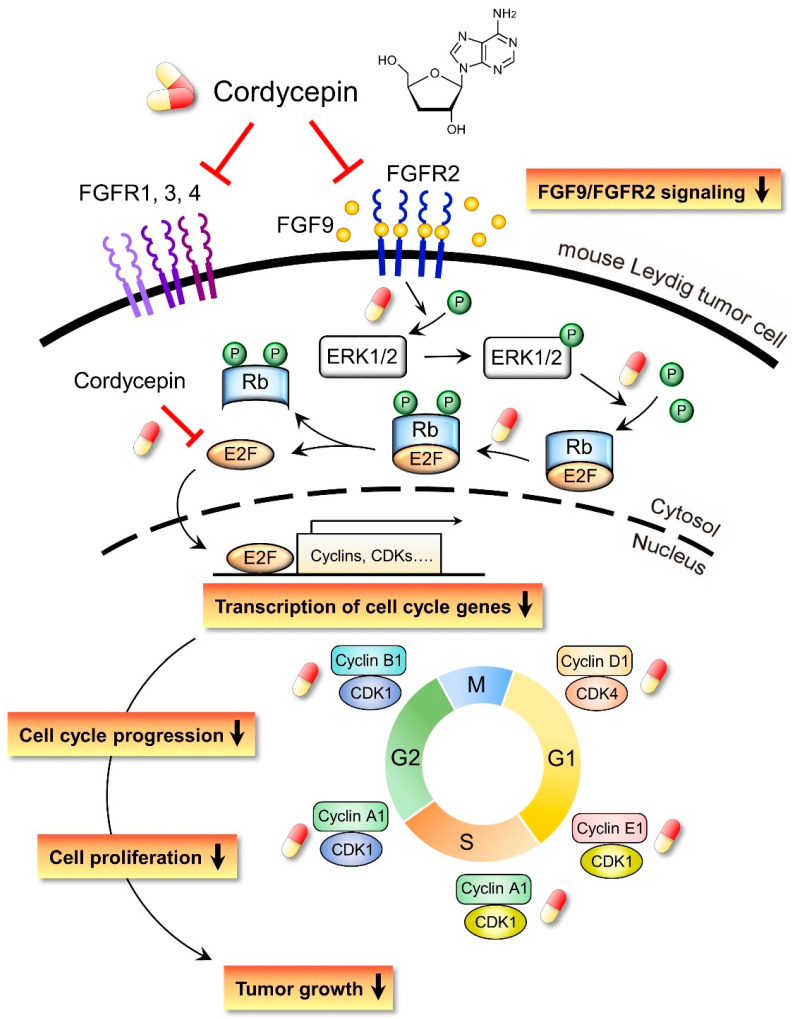
Schematic representation of the anticancer effects of cordycepin on FGF9-induced mouse Leydig tumor cells. Cordycepin inhibited FGF9-induced MA-10 cell proliferation in vitro and tumor growth in vivo by suppressing the expression of p-ERK1/2, p-Rb, E2F1, cyclins, CDKs and FGFR1-R4 proteins. The red and yellow capsule symbol means the FGF9-induced expression of protein was inhibited by cordycepin.

**Table 1 ijms-21-08336-t001:** The fold change of the mRNA levels of FGFs and FGFRs regulated by cordycepin vs. control at 3 hours of treatment in MA-10 cells.

Gene Symbol	Cordycepin/Control	
	Log_2_ (ratio)	*p*-Value
FGF9	−2.162	0.016
FGF18	−1.124	0.009
FGFR1	0.429	0.694
FGFR2	−1.250	0.0003
FGFR3	−0.614	0.023
FGFR4	−0.110	0.905

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
