# Peer review of "Anti-Cancer Effect of Cordycepin on FGF9-Induced Testicular Tumorigenesis"

_ijms, 2020, doi:10.3390/ijms21218336_

Round 1

Reviewer 1 Report

In this manuscript, Chang et al. demonstrates that Cordycepin affects the growth and viability of MA-10 leydig cell tumor via negative regulation of the expression and the activity of FGF9-signalling pathway.

The authors first introduce the Fgf9 and 18 as down-regulated transcripts in their previous microarray result. The effect of ectopic Fgf9 on the activation of MAP kinase pathway as well as cell-cycle regulators, and cancellation of this effect by Cordycepin is demonstrated. This cancellation is explained by the down regulation of FGFR expression in the following figures. Finally the effect of ectopic FGF9 signalling on enhancing tumor growth in vivo using xenograft assay and its cancellation by Cordycepin is examined.

The experiments are well executed and the manuscript is well written, and the finding that Fgf9 is a target of Cordycepin mediated cell growth in MA-10 leydig cell tumor is potentially interesting. On the other hand, as Cordycepin affects the cells in various means and decreases viability and induces apoptosis of MA-10 cells by 24 hours under 100uM (as documented in the authors’ previous publications: Ref# Sci. Reports 2015), there are some concerns regarding direct versus indirect effects of Cordycepin to FGF9 signalling pathway, and its physiological relevance for the MA-10 leydig cell tumor growth.

Major comments:

One major concern regarding the project is the relative broad effect of Cordycepin in tumor cells, as the authors previously published. As such, the relevance of the inhibiting FGF9-signalling pathway for the inhibition of MA-10 tumor growth is not firm enough at this point.

The authors show that Fgf9 is a target of Cordycepin, by microarray analysis. This implies that at least in culture, FGF9 can function in an autocrine manner. If this pathway is important as the authors claim in the title, inhibition of FGF9 (by knock-down approach) or inhibition of the receptor itself may have similar consequences both in culture and in vivo. Can the authors demonstrate if this is the case?

Minor comments

  1. Quantification of the signals in tumors (Figure 7-9.) simply by image analysis is not robust enough. Can the authors demonstrate this is the case through other approaches such as Western blot?
  2. There are some typos in the text (i.e. Figure 8 legend title Coroycepin -> Cordycepin etc.). The authors should check carefully before publication.

Reviewer 2 Report

The authors reported that cordycepin-treated Leydig tumor cells from MA-10 mouse were shown down-regulated the mRNA levels of FGF9, FGF18, FGFR2 and FGFR3 genes. This study is expected to improve the value of cordycepin produced from Cordyceps and to present new possibility as an anticancer drug for testicular cancers. The objectives and methods of the study are carefully presented, and the results support the aim of the work. In particular, schematic representation of the anticancer effect of cordycepin in Figure 10 well summarizes the results and importance of this study, thus, it is expected to be an attractive guideline for subsequent studies.

Therefore, it is recommended that “Accept after minor revision”

Minor points

1) There are some syntax errors throughout the manuscript, and the writing style is not appropriate as a scientific paper. Professional English editing services are highly recommended.

2) Italic errors are found in the text of this manuscript, check these points carefully.

 Line: 21, 22, 51, 55, 56, 125, 145, 146, 368, 370, 538~

3) Since the resolution of the Figures are poor, thus, it is recommended to provide Figures at a higher resolution.

4) References should be carefully revised as the journals format.

Journal names should be indicated in italic and abbreviation followed by comma.

Typos

Line 44:  Leydig_cell  ----à Leydig cell

Line 47: ~ toxicity. ( ‘.’ Is red color) ---à change to black

Line 58: ~ metastasize. ( ‘.’ Is red color) ---à change to black

Line 610: subcutaneous (sc) ---à subcutaneous (SC)

Line 614: 100  μl of 50 ng/ml FGF9 or 0.0025 % (It is recommended to rewrite sentences with correct expressions.)

Line 620: reference 51

Round 2

Reviewer 1 Report

In the authors response, the authors mention that they have previously shown that FGF9 mediated-FGFR activation stimulates MA-10 cell growth (Ref31). However, the effect of inhibiting this pathway in MA-10 cell growth has not been addressed properly (the authors have only shown that FGFR inhibition negatively affects the ectopic FGF9-stimulated growth in culture).

As mentioned in the initial comment, Cordycepin affects the cells in various ways and decreases viability and induces apoptosis of MA-10 cells. Therefore, the following concerns make the significance of this work questionable.

(i) Direct versus indirect effects of Cordycepin to FGF9 signalling pathway (could it be just unhealthy cells expressing lower levels of transcripts that are important for cell growth?)

(ii) Relevance of the inhibition of this pathway for Cordycepin mediated MA-10 growth viability (could it be just unhealthy cells not responding to FGF9).